# Sirtuin 6 Overexpression Improves Rotator Cuff Tendon-to-Bone Healing in the Aged

**DOI:** 10.3390/cells12162035

**Published:** 2023-08-10

**Authors:** Young Jae Moon, Baoning Cui, Se-Young Cho, Jae Won Hwang, Hee-Chung Chung, Joseph Kwon, Duwoon Kim, Kyu Yun Jang, Jung Ryul Kim, Sung Il Wang

**Affiliations:** 1Department of Biochemistry and Orthopaedic Surgery, Research Institute of Clinical Medicine of Jeonbuk National University-Biomedical Research Institute of Jeonbuk National University Hospital, Jeonbuk National University Medical School, Jeonju 54896, Republic of Korea; 2Department of Orthopaedic Surgery, Research Institute of Clinical Medicine of Jeonbuk National University-Biomedical Research Institute of Jeonbuk National University Hospital, Jeonbuk National University Medical School, Jeonju 54896, Republic of Korea; c30392@outlook.com (B.C.);; 3Department of Food Science and Technology, Foodborne Virus Research Center, Chonnam National University, Gwangju 61186, Republic of Korea; 4Department of BioChemical Analysis, Korea Basic Science Institute, Daejeon 30147, Republic of Korea; 5Department of Pathology, Jeonbuk National University Medical School, Jeonju 54896, Republic of Korea

**Keywords:** rotator cuff, recurrent tear, SIRT6, tendon-to-bone healing

## Abstract

Aging is an independent risk factor for recurrent tearing after surgical repair of rotator cuff ruptures around the tendon-to-bone area. However, aging signature factors and related mechanisms involved in the healing of the rotator cuff are still unknown. We hypothesized that differences in proteins involved in the rotator cuff according to age may affect tendon-to-bone healing. The proteome analysis performed to identify the signature aging proteins of the rotator cuff confirmed the sirtuin signal as an age-specific protein. In particular, the expression of SIRT6 was markedly down-regulated with age. Ingenuity pathway analysis of omics data from age-dependent rat rotator cuffs and linear regression from human rotator cuffs showed SIRT6 to be closely related to the Wnt/β-catenin signal. We confirmed that overexpression of SIRT6 in the rotator cuff and primary tenocyte regulated canonical Wnt signaling by inhibiting the transcriptional expression of sclerostin, a Wnt antagonist. Finally, SIRT6 overexpression promoted tendon-to-bone healing after tenotomy with reconstruction in elderly rats. This approach is considered an effective treatment method for recovery from recurrent rotator cuff tears, which frequently occur in the elderly.

## 1. Introduction

The enthesis of the rotator cuff, which acts to stabilize and allow extensive motion of the shoulder, connects the non-mineralized tendon and mineralized humeral head. The enthesis experiences an accumulation of high stresses based on its location between those two types of tissues and its own tissue inhomogeneity, resulting in a high rate of tear and delayed healing [1]. Delayed tendon-to-bone healing after surgical intervention due to rotator cuff tear (RCT) potentially causes rotator cuff repair failure and requires revision of the primary repair or extensive surgery such as tendon transfer, grafts for reconstruction, or total reverse shoulder arthroplasty [1,2].

The biological processes of tendon-to-bone healing consist of inflammation, proliferation, matrix synthesis, and remodeling [3]. Initially, hematomas are formed, growth factors are released, and infiltrated inflammatory cells interact with resident stem cells and bone-marrow-derived stem cells around the injured area [4]. Next, stem cells proliferate and differentiate into mineralized chondrocytes, tendon sheath cells, and osteocytes based on biological signals [5]. The differentiated matrix-producing cells and recruited fibroblasts synthesize the extracellular matrix (ECM), and the newly formed bone and interfacial tissue are re-established. Based on this biological healing process, animal studies have attempted to enhance the local healing environment and improve tendon-to-bone healing by adding growth factors, bone-marrow-derived stem cells [6], or platelet-rich plasma [7] or by inhibiting metalloproteinases [8].

The prevalence of RCT is approximately 50% in adults over the age of 70 years. Despite the improvement in the mechanical strength of rotator cuff repairs, re-tear rates vary between 20% and 40% [9]. Advanced patient age is an independent risk factor of recurrent tearing after rotator cuff repair surgery [9,10]. Some research studies suggested that the impaired biological environment and diminished tendon-to-bone healing are related to poor outcomes in older patients with RCT [11,12]. Plate et al. reported diminished tendon-to-bone healing after rotator cuff repair in a rat model of aging [13]. However, the underlying mechanisms and associated molecular mechanisms in the increasing incidence of recurrent RCT that are related to aging have not yet been elucidated.

The tendon-to-bone interface consists of several ECM proteins, including collagens. These tendon-to-bone interface constructs of the rotator cuff have been reported to differ in structure and composition with age. Histologic analysis shows an enlargement of the mineralized area, reduction in the non-mineralized area, and cellular disorganization in the aged rat rotator cuff [14]. Proteomics analysis shows markedly different ECM proteins between newborn (day 0) and young (2–3 months) rats, and mild changes in adult (6–8 months) and old (15–18 months) rats from the rotator cuff [15]. Because the ECM regulates cell phenotype and function through cell-matrix signaling in tissue homeostasis and in response to injury [16], ECM changes with age may influence tenocytes and fibrochondrocytes, which are major cells in the tendon-to-bone interface. However, biomolecules related to changes in the rotator cuff according to age have not been identified. We hypothesized that there would be biomolecular signaling related to the age-dependent proteome in the rotator cuff and analyzed the proteome from the rotator cuff with age using bioinformatics.

Sirtuins are highly conserved class III nicotinamide adenine dinucleotide (NAD)-dependent deacetylases that target histone and nonhistone proteins [17]. Recent studies indicate their involvement in a variety of functions ranging from longevity, genomic stability, tumorigenesis, and inflammation to metabolic diseases through the deacetylation function for transcriptional regulation and protein stability [18,19]. Sirtuin 6 (SIRT6) is a member of the sirtuin family of NAD-dependent deacetylases. SIRT6 regulates lifespan and several fundamental processes controlling aging, such as metabolism, telomere maintenance, and DNA repair [20,21,22,23]. In particular, overexpression of SIRT6 suppresses inflammatory diseases such as rheumatoid arthritis [24] and ischemic osteonecrosis [25] and promotes wound healing [26]. Thus, the sirtuin pathways are an attractive therapeutic approach for regenerative medicine in age-related tissue repair.

Wnt/β-catenin signaling has a major role in tendon formation at embryogenic development. The depletion of ectodermal Wntless, which is required for the secretion of various Wnts, is observed as dysgenesis of the tendon, and the loss of mesenchymal β-catenin results in the impairment of tendon formation [27]. From the point of view of tendon regeneration, Wnt activation using the sclerostin (Wnt antagonist) neutralization antibody improves tendon-to-bone healing in the rotator cuff acute injury mice model [28]. In addition, emerging studies have introduced Wnt antagonists as aging signature proteins that increase with age [29,30,31]. Those pieces of evidence strongly imply that canonical Wnt signaling is necessary for tendon morphogenesis and regeneration in aging.

Here, we showed that age-related changes in tendon-to-bone proteins were related to the sirtuin signaling pathway. Notably, SIRT6 decreased significantly with age in both rats and human subjects. This change resulted in Wnt inhibition through the regulation of sclerostin. However, the activation of Wnt through SIRT6 overexpression promoted tendon-to-bone healing in aged rats.

## 2. Materials and Methods

### 2.1. Patients

Human rotator cuffs were acquired from a nail insertion site during the antegrade intramedullary nailing from patients diagnosed with humerus fracture of the proximal and middle thirds without evident RCT from March 2017 to December 2020. The obtained cuff tissues were stored in PBS solution at −80 °C at our Human Resources Bank. A written form of informed consent was obtained at the time of surgery from all patients. To investigate the protein and mRNA levels of SIRT6 according to age, rotator cuff samples from patients in their 20 s to 30 s (*n* = 4), 50 s to 60 s (*n* = 6), and 70 s to 80 s (*n* = 6) were analyzed. And to confirm the correlation between SIRT6 and β-catenin, rotator cuff samples in their 50 s and 60 s (*n* = 13) were analyzed. The study was conducted according to the guidelines of the Declaration of Helsinki and approved by the Institutional Review Board (IRB No. 2017-02-020).

### 2.2. Study Design and Surgical Technique

To determine age-dependent differences in tendon-to-bone healing, age-matched male Sprague Dawley rats (young: 2-month-old, 200–300 g; adult: 7-month-old, 400–500 g; and aged: 15-month-old, 600–800 g, each group used nine rats, Jung-Ang Lab Animal Inc. Seoul, Republic of Korea) underwent right side supraspinatus tendon transection and acute repair, similar to previous reports [6,7]. For sham surgery, a split incision and skin suturing of the left side deltoid muscle of the same rat was performed. Each group used nine rats. The rats were anesthetized with an intraperitoneal injection of ketamine (80 mg/kg; Fort Dodge Animal Health, Parsippany-Troy Hills, NJ, USA) and xylazine (5 mg/kg; Akorn Inc., Lake Forest, IL, USA). All operations were performed by use of a sterile technique with the rat in the lateral decubitus position. A deltoid-splitting incision was made, and the acromioclavicular joint was divided, allowing visualization of the rotator cuff tendons (Figure 1A). The supraspinatus tendon was isolated, and a modified Mason–Allen stitch was placed by use of a 4-0 Ethibond (Johnson & Johnson Inc., Guelph, ON, Canada) nonabsorbable suture. The tendon was then sharply detached from the greater tuberosity and the footprint gently decorticated with a scalpel blade to ensure complete debridement of the native enthesis (Figure 1B). Crossed bone tunnels were drilled at the anterior and posterior margins of the footprint and 2 mm lateral to the articular surface by use of a 22-gauge needle (Becton Dickinson, Franklin Lakes, NJ, USA) (Figure 1C). Suture ends were then passed through the bone tunnels and firmly tied over the humeral metaphyseal cortex, anatomically repairing the supraspinatus tendon to its native footprint (Figure 1D) [32]. The deltoid split and wound were subsequently closed in a standard layered fashion with absorbable sutures. At 4 weeks after surgery, shoulder specimens were subjected to biomechanical testing (*n* = 5) and histological assay (*n* = 4) to compare among the age groups. Finally, we performed the SIRT6 overexpression experiment using adenovirus SIRT6 (Ad-SIRT6) and β-galactosidase (Ad-LacZ) in 15-month-old rats (Figure 1E) and evaluated the experiment using biomechanical testing (*n* = 5) and a histological assay (*n* = 4). Sham surgery was performed through a deltoid-splitting incision and skin suture, and eight animals in each group (2-month-old, 7-month-old, and 15-month-old) were used to compare the protein and mRNA expression of rotator cuffs according to age. The animal study protocol was approved by the Institutional Animal Care and Use Committee (permit no. 2017–0111).

### 2.3. Histology

Rotator cuffs with a humeral head were immediately placed in 10% formalin solution and decalcified in 10% EDTA for 2 months. Tissue sections were collected from the midline of the supraspinatus. For histologic findings, paraffin-embedded tissue sections (5 μm) were stained with hematoxylin and eosin (H&E). Masson’s trichrome staining was performed with a commercial kit (Abcam, Cambridge, UK). Safranin O staining was performed using Fast Green and Safranin O. Immunohistochemistry was performed using a specific HRP/DAP detection IHC kit (Abcam). Images were acquired using a Leica DM750 microscope (Leica, Wetzlar, Germany). The area of interest to measure for histological analysis was the tendon-to-bone insertion (enthesis) site. The area was measured using the iSolution DT 36 software (Carl Zeiss, Oberkochen, Germany).

### 2.4. Proteome Analysis on Rotator Cuffs

Four rotator cuffs from 2-month-old and 15-month-old rats were isolated, and proteins were extracted. The protein concentrations were quantified using BCA protein assay kit (Thermo, Waltham, MA, USA, 23227). After the quantification, the protein solutions were made of tube gel and were chopped to 1~2 mm^2^ in size. After the gels were completely dried in speed vac, the gels were digested using trypsin (1 µg) in 100 μL with 50 mM ABC at 37 °C for 16 h. The peptides were collected in the gel using each solvent. The desalted peptide mixtures (100 μL of 0.1% formic acid, 3% ACN) were analyzed in AcquityTM HSS T3 1.8 μm TrizaicTM nano-Tile column (85 μm × 100 mm) using a nano-ACQUITY Ultra performanceTM chromatography system (Waters Corporation, Milford, MA, USA) with a synapt G2-Si HDMS system (Waters Corporation, Milford, MA, USA). MS spectral data were collected in triplicate. The 4 μL volume of tryptic peptides were injected by the auto sampler. [Glu1]-fibrinopeptide (500 nL /min) was used to calibrate the time-of-flight analyzer in the range of *m*/*z* 100−1500, and [Glu1]-fibrinopeptide (*m*/*z* 785.8426) was used for lock mass correction. The MS raw data were processed using the Progenesis QI for Proteomics (QIP) version 2.0 (Nonlinear Dynamics, UK). The protein database was analyzed using rattus norvegicus (Rat) (Uniprot: UP000002494, Accessed: 01/22/19). And the database search was conducted used the following parameters: lock mass calibration, 785.8426; low-energy threshold, 150 counts; elevated energy threshold, 50 counts (MSE parameters); digest reagent, trypsin; peptide false discovery rate (FDR), 4%; fragments/peptide, 3 or more; fragments/protein, 7 or more; and peptides/protein, 1 or more (search parameters). Protein-level relative quantitation was performed using the Hi-N approach (*n* = 3) as implemented in the QIP [33].

### 2.5. Biomechanical Test

Each dissected supraspinatus with humerus was installed in a universal testing machine (Model 4201, Instron, Canton, MA, USA) to measure the tensile strength between the supraspinatus and the proximal humerus. The bonded specimens were fixed to the jig having a mini-dumbbell shape, and a tensile force was loaded at a crosshead speed of 10 mm/min. The computer connected to the machine produced the load-extension curve. Maximum force at rupture (kgf) and extension (mm) were evaluated from the curve.

### 2.6. Bioinformatics Analysis

The quantitative data from 86 proteins were imported into Ingenuity^®^ Pathway Analysis (IPA) software version 2.3 (QIAGEN Inc., Hilden, Germany, https://digitalinsights.qiagen.com/IPA, accessed on 8 July 2019) and were analyzed through canonical pathway analysis, upstream regulators, and molecular networks. The statistical significance of each pathway was determined through IPA using a Fisher Exact test *p* < 0.05 [34].

### 2.7. Quantitative Real-Time PCR

Total RNA was extracted with TRizol reagent (Invitrogen, Life Technologies, Carlsbad, CA, USA) according to the manufacturer’s method. cDNA was synthesized using first-strand cDNA synthesis kits according to the manufacturer’s method (Applied Biosystems, Foster City, CA, USA). Quantitative polymerase chain reaction (PCR) was performed in 384-well plates using an ABI Prism 7900HT Sequence Detection System (Applied Biosystems). Relative amounts of mRNA were quantified using the comparative Ct method (ΔΔCt). All mRNA values were normalized to those of *GAPDH, Actb,* or *Rps3*. Primers used in PCR are listed in Appendix A.

### 2.8. Western Blot

Tissue and primary cells were homogenized in Mammalian Protein Extraction Reagent (Thermo Fisher Scientific, Waltham, MA, USA). Protein lysates (20 μg) were separated using 10% SDS-PAGE and transferred to nitrocellulose membranes. After blocking with 5% skim milk for one hour, the blots were probed with primary antibodies. Antibodies were used against the following proteins: SIRT1 (ab110304) and SIRT5 (ab195436, Abcam, Cambridge, UK); SIRT2 (sc-28298) and β-actin (sc-47778, Santa Cruz Biotechnology, Dallas, TX, USA); β-catenin (71-2700, Themo Scientific, Waltham, MA, USA); SIRT4 (3224) and SIRT7 (3099, Biovision, Cambridge, UK); active β-catenin (8814s), cyclin D1 (55506s), SIRT3 (5490s), and SIRT6 (12486s, Cell Signaling Technology, Danvers, MA, USA); GAPDH (AP0066, Bioworld Technology, St. Louis Park, MN, USA); DKK1 (21112-1-AP; Proteintech, Sankt Leon-Rot, Germany); and SOST (bs-10200R, Bioss, Woburn, MA, USA). Antibody signals were detected using an LAS-3000 Luminoimage analyzer (Fujifilm, Tokyo, Japan). ImageJ software version 1.4 (NIH, Bethesda, MD, USA) was used for densitometry of the gel images of protein bands for patient samples. All protein intensities were normalized to those of GAPDH.

### 2.9. Primary Culture of Tenocytes, Infection Using Adenovirus, and Transfection

The supraspinatus tendons were isolated from aged (15-month-old) rats. Tendons were minced using scissors in Hank’s Balanced Salt Solution (HBSS, Invitrogen) and enzymatically digested at 37 °C using collagenase (2 mg/mL, Clostridopeptidase A, Sigma, Saint Louis, MO, USA) diluted in D-MEM (Dulbecco’s Modified Eagle Medium). The digested product was centrifuged, and the pellet was re-suspended and cultured in D-MEM with 10% fetal bovine serum, 1% penicillin/streptomycin, and 2% L-glutamine. Cultured primary tenocytes were overexpressed with SIRT6 using adenovirus. Adenoviruses expressing SIRT6 (Ad-SIRT6), a catalytically inactive mutant SIRT6 (Ad-mtSIRT6), or β-galactosidase (Ad-LacZ) were prepared as described previously [25]. For knockdown of SOST, primary cells were transfected with 30 pmol of siRNA targeting for *Sost* (Bioneer, Daejeon, Republic of Korea). The siRNA of the *Sost* duplex had the sense and antisense sequences 5′-GACAACAACCAGACCAUGAGACUCAGAGAGUACCCAGACCUCAGGAACUAGAGAACA-3′ and 5′-UCAUGGUCUGGUUGUUCUCUCUGGGUACUCUCUGAGUCUGUUCUCUAGUUCCUGAGG-3′, respectively, using Lipofectamine RNAiMAX Reagent (Thermo Fisher Scientific, Waltham, MA, USA). For overexpression of SIRT6 or SOST, cells were transfected using Lipofectamine 3000 (Invitrogen, Carlsbad, CA, USA) with 1 μg of plasmid expressing SIRT6. A pFLAG-CMV-2 plasmid vector was used as a control vector. The vector overexpressing wild-type SIRT6 was synthesized by Cosmogenetech Co., Ltd. (Cosmogenetech Co., Ltd., Seoul, Republic of Korea). The vector overexpressing SOST was synthesized by OriGene Technologies, Inc. (Ori-Gene Technologies, Inc., Rockville, MA, USA).

### 2.10. Statistical Analysis

Data are expressed as the mean ± standard error of the mean. SPSS software version 25.0 (IBM, Armonk, NY, USA) was used to perform the statistical analyses. Comparisons between two groups were evaluated using an unpaired two-tailed Student’s *t* test. To compare more than two groups, we used one-way ANOVA followed by Tukey’s test for multiple comparisons. A *p* value less than 0.05 was considered significant.

## 3. Results

### 3.1. Tendon-to-Bone Healing Is Impaired in the Rotator Cuff Injury of Aged Rats

To identify the change in rotator cuff with age, we observed the supraspinatus tendon insertion site of the humeral head of aged rats. Most of the supraspinatus tendon insertion sites in 2-month-old rats consisted of collagen tissue stained blue on Masson’s trichrome staining (Figure 2A). However, with age, the distribution of red-colored elastic fibers increased instead of decreased the collagenous fibers around the tendon-to-bone lesions (Figure 2A). To verify that age-related changes in tendon-to-bone composition affect recovery after acute injury, we repaired the injured rotator cuff and evaluated the degree of recovery after four weeks. When comparing the mechanical strength of the suture site after rotator cuff tenotomy (right) and sham surgery site (left) in the same rats according to age, the maximum load was significantly reduced at the repair site compared to the contralateral sham-surgery site in 15-month-old rats. This difference in right and left maximal load was observed considerably in 15-month-old rats compared to 2-month-old rats (Figure 2B). In addition, the new fibrocartilage formation area (the metachromasia area), which is stained red in safranin O staining after tendon repair surgery and represents tendon-to-bone healing [35], was suppressed in the aged group compared to the young group (Figure 2C). These results suggest that the properties of the rotator cuff, which change with age, may affect tendon-to-bone recovery after rotator cuff injury.

### 3.2. Expression of SIRT6 Is Down-Regulated with Age in the Rotator Cuff

To confirm which proteins and related pathways were differentially regulated across the age in the rotator cuff, we analyzed the quantification of protein expression according to age in the rotator cuff of rats (2-month-old vs. 15-month-old). We found 86 proteins that significantly changed with age and bioinformatically determined annotation terms that were statistically enriched in the group (Appendix A). Interestingly, this revealed that glucose metabolism such as glycolysis (−log(*p*-value) = 16.3) and gluconeogenesis (−log(*p*-value) = 14), calcium-related signal (−log(*p*-value) = 9.02), and sirtuin signaling pathways (−log(*p*-value) = 5.42) were the most enriched pathways in the young rotator cuff compared with the old-aged rotator cuff (Figure 3A). Because sirtuins are essential factors that delay cellular senescence and extend the lifespan [19], we focused on the sirtuin pathway. The volcano plot indicated six proteins related to the sirtuin signaling pathway as purple circles (Figure 3B, Appendix A). We confirmed that these proteins are significantly expressed between different age groups. Next, we evaluated the expression of sirtuins in the rotator cuffs of rats according to age. With increasing age, the expression of SIRT1 increased while that of SIRT6 markedly decreased (Figure 3C). Tenocytes and fibrochondrocytes expressed SIRT6 in the nucleus at 2 months old. However, those expressions and that of mRNA *Sirt6* decreased with age (Figure 3D–F). Furthermore, the SIRT6 protein and its transcriptional level in the rotator cuff of human subjects were also significantly down-regulated with age (Figure 3G–I).

### 3.3. Wnt/β-Catenin Signaling Is Suppressed with Increasing Age in the Rotator Cuff

To understand the mechanisms between differentially expressed proteins and proteins related to the sirtuin signaling pathway, molecular network analysis was performed based on the Ingenuity Pathways Knowledge Database. The CTNNB1 gene encoding the β-catenin protein was predicted to be activated at the young age group by upstream regulator analysis of IPA (Z-score = 1; *p*-value of overlap = 4.08 × 10^−3^), suggesting that the expression of the CTNNB1 gene is inhibited at an old age group by regulating the expression of proteins including SERPINA1, BGN, CA3, and ENO3 (Appendix A). Interestingly, the protein expressions of SIRT6 in each human rotator cuff were significantly positively correlated with the protein expressions of active β-catenin, which is a key downstream effector in the Wnt signaling pathway (Figure 4A and Appendix A). Consistent with the results from human subjects, active β-catenin in the rat rotator cuff decreased similarly to SIRT6 expression, which was attenuated with age (Figure 3C and Figure 4B). Although the transcriptional level of β-catenin did not decrease significantly, the mRNA level of cyclin D1, the target gene of β-catenin, decreased significantly with age (Figure 4C). Furthermore, similar to SIRT6 expression, tenocytes and fibrochondrocytes expressed β-catenin in the nucleus of 2-month-old rats. However, those expressions were attenuated in old age (Figure 4D).

### 3.4. SIRT6 Modulates Sclerostin, Which Is an Antagonist of the Wnt Signaling Pathway

Since the change in the transcriptional level was not significant compared to the expression level of β-catenin protein with age, it was confirmed whether SIRT6 regulates β-catenin in the nucleus through protein–protein interaction. We transfected HEK293T cells with β-catenin and SIRT6, but compared to the control vector, there was no change in the nuclear expression of β-catenin, indicating that SIRT6 had no controllability through direct binding to β-catenin in the nucleus (Appendix A). Next, we sought to verify whether SIRT6 regulates the expression of antagonists (SOST, DKK1) that inhibit Wnt/β-catenin signaling because SIRT6 deacetylates histone protein, thereby serving as a repressor of gene expression. In addition, in recent studies, since the Wnt antagonist suppresses the Wnt signal due to increased expression in the elderly [36], we confirmed whether the same phenomenon occurred in the rotator cuff. We checked the expression of β-catenin and its antagonists over 2 weeks after injecting adenovirus SIRT6 into the rotator cuff of the old rats. The active β-catenin and its target protein (cyclin D1) were increased by overexpressing SIRT6 in the rotator cuff. Conversely, the protein expressions of SOST and DKK1, antagonists for Wnt signaling, were suppressed (Figure 5A). To reconfirm whether these results occur in tenocytes, we performed a primary culture of tenocytes from the rotator cuff of old rats, then infected them with SIRT6 and checked the antagonists. In primary tenocytes, both the protein and mRNA levels of SOST were suppressed by SIRT6, but DKK1 remained unchanged, suggesting that SIRT6 activates Wnt signaling by regulating SOST expression transcriptionally in tenocytes (Figure 5B,C). To identify a more detailed mechanism, we performed an inhibition SOST or an overexpression SOST experiment with overexpressed SIRT6. As expected, active β-catenin was upregulated by the knock-down of SOST and was downregulated by the overexpression of SOST in primary tenocytes from the aged rat (Figure 5D–G). Consistent with the SIRT6 overexpression using adenovirus, transfected SIRT6 using the plasmid upregulated active β-catenin to a level similar to that of the SOST knock-down group. The expression of active β-catenin upon knock-down of SOST in SIRT6 overexpression did not increase compared to the SIRT6 overexpression group or the SOST knock-down group but was similar (Figure 5D,E). In addition, the increased expression of active β-catenin by SIRT6 overexpression was significantly downregulated by overexpression of SOST (Figure 5F,G). These results demonstrate that the regulation of β-catenin expression by SIRT6 is not directly regulated by SIRT6 but through SOST.

### 3.5. SIRT6 Overexpression Enhances Tendon-to-Bone Healing in Acute Rotator Cuff Injury in Aged Rats

Next, to confirm the effect of SIRT6 on tendon-to-bone healing in the aged, we performed acute injury and repair of the rotator cuff in aged rats and infected the repair site with adenovirus SIRT6. The SIRT6-overexpressing group significantly restored the tensile strength of the supraspinatus tendon (Figure 6A). Furthermore, the metachromasia area was enlarged in the SIRT6-overexpressing aged group compared to the control group (Figure 6B). Consistent with the results of primary tenocytes, overexpression of SIRT6 increased the nuclear expression of β-catenin in the fibrochondrocytes region (Figure 6C).

## 4. Discussion

RCT is one of the most common causes of pain and dysfunction in the shoulder. Although the etiology of RCT has not yet been clarified, the alternation of collagen fibers and age-related changes are accepted intrinsic factors [37,38]. Meanwhile, in spite of the significant improvement in the surgical techniques, the rate of re-tear is still high, and several studies report old age, tear size, and tissue quality as important risk factors for re-tear [1,2]. These suggest that in the elderly, there are changes in the proteins constituting the rotator cuff and signals related to regeneration compared to the young. The impaired biological environment and diminished tendon-to-bone healing lead to the poor outcomes seen in older patients [11,12]. In the present study, we established that proteins constituting the rotator cuff, which change with age, were associated with the sirtuin signal pathway. In particular, SIRT6 was significantly attenuated in the rotator cuff of both rats and humans, and in this regard, we demonstrated that SIRT6 regulated canonical Wnt signaling. Finally, we confirmed that overexpression of SIRT6 improved the Wnt signaling activity and tendon-to-bone healing around reconstructed tissues after tendon injury in aged rats. Although the tendon-to-bone healing results are from studies in animal models, these data suggest the possibility of improving the healing rate and clinical outcomes of aged RCT with SIRT6 activators.

In the tendons of the old men, the concentration of collagen was reduced and that of elastic fibers, which accumulate advanced glycation end products, was increased compared to the young [39]. In the current study, the amount of collagen in the rat rotator cuff similarly decreased with age, and the elastic fibers also increased. These changes increased the tensile strength in aged rats but rather worsened tendon-to-bone healing after injury and repair. We can infer the relationship between changes in collagen and tendon-to-bone healing through the sirtuin pathways identified using bioinformatics. Meanwhile, among the sirtuins, SIRT6 contributes to collagen synthesis and degradation by regulating NF-κB signaling, a significant signal of inflammation [40]. In addition, SIRT6 promotes wound healing and plays a pivotal role in collagen deposition by regulating the polarization of M1 to M2 macrophages [26]. In our study, the expression of SIRT6 with those roles decreased with age in the rotator cuff of humans and rats, which may partially explain why aging may attenuate collagen deposition and interfere with tendon-to-bone healing.

We showed that the Wnt-β-catenin signaling pathway was another pathway that affects tendon-to-bone healing with age. With age, the expression of β-catenin in the rotator cuff is attenuated, and we confirmed that the expression of β-catenin increased when tendon-to-bone healing was improved. According to previous studies, the loss of function of β-catenin in mesenchymal cells leads to the impairment of tendon formation and suppression of the expression of Dermo1 and Scleraxis, two significant genes for tendon formation [29]. Furthermore, Wnt-β-catenin activation promotes tendon-to-bone healing in a rodent model of rotator cuff acute injury [28,41,42]. Those results suggested that Wnt signaling is an essential mechanism of action for tendon-to-bone healing. In the present study, we confirmed that the Wnt-β-catenin signaling pathway, which plays this role in the tendon, was regulated by SIRT6. Firstly, the age-dependent rotator cuff proteomics results through IPA showed that the Wnt-β-catenin and sirtuin signaling pathways were significantly related. Second, we verified that the protein expression level of SIRT6 in the human rotator cuff significantly correlated positively with that of active β-catenin. Third, overexpression of SIRT6 using adenovirus in the rotator cuff of aged rats resulted in the enhancement of active β-catenin. Lastly, the overexpression of SIRT6 in primary tenocytes significantly reduced the protein and transcriptional expression of sclerostin, which is a Wnt antagonist. This relationship between sirtuin and Wnt signaling is supported by other studies involving sirtuins. Increased SIRT1-dependent deacetylation by deleting an oxygen sensor protein in osteocytes directly acts on the *Sost* promoter to reduce sclerostin expression and increase Wnt-β-catenin signals to increase bone mass [43]. A deficiency in SIRT6 causes the hyperacetylation of H3K9 in the *Dkk1* promoter, resulting in osteopenia [44]. Similar to the present study, sirtuin has been reported as an epigenetic silencing-targeting Wnt antagonist. Other studies reported that SIRT1 [45] or SIRT6 [46] interacts with β-catenin to induce the nuclear accumulation of β-catenin and regulate cell proliferation and differentiation through the activation of Wnt signaling.

Sclerostin (SOST) has been identified as binding to LRP5/6 receptors and inhibiting the Wnt signaling pathway [47]. More recently, sclerostin has been identified as an aging signature protein by analyzing the human plasma proteome [29,30,31]. Our results showed that the overexpression of SIRT6 in the rotator cuff of aged rats reduced the protein level of the aging protein sclerostin. This regulatory mechanism confirmed that SIRT6 directly suppressed the transcriptome of sclerostin on tenocytes. However, since sclerostin is highly expressed in bone and cartilage [48], the possibility of suppressing the expression of sclerostin by affecting the surrounding tissues (bone, cartilage) of the rotator cuff cannot be excluded.

Our study does have some limitations. It is challenging to apply the methodology of SIRT6 overexpression using the virus to the human rotator cuff due to safety, transduction efficiency, immunogenicity, and target cell specificity [49]. Therefore, it is necessary to study drugs or other methods that can activate SIRT6. Recently, a drug that activates SIRT6 has been discovered and applied to several mouse disease models [50,51]. We plan to use this drug in a rotator cuff injury model in aging mice. In addition, the expression of SIRT1, which has a similar role to SIRT6, increased with age in the rotator cuff, unlike SIRT6. This result was thought to increase the elastic fibers constituting the tendon, but more research is needed to confirm the precise function of SIRT1.

## 5. Conclusions

In summary, we find that the proteins that make up the rotator cuff change with age, and these changes affect tendon-to-bone healing. We demonstrate SIRT6 as an aging-associated protein in the rotator cuff and found that it regulates Wnt signaling. We further show that SIRT6 overexpression improves tendon-to-bone healing after acute injury in the aged rotator cuff. Our data suggest the therapeutic potential for recovery from tears of aged rotator cuffs through the SIRT6 activator.

## Figures and Tables

**Figure 1 cells-12-02035-f001:**
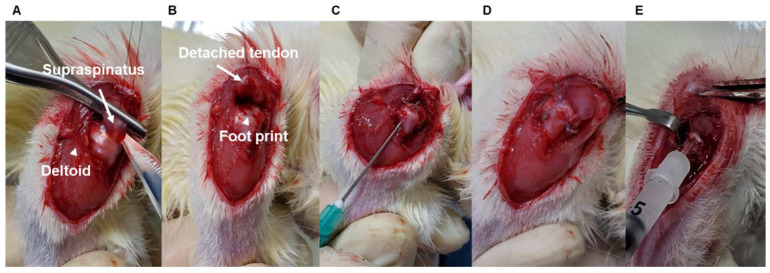
Representative pictures of the surgery process. (**A**) The supraspinatus tendon (white arrow) shown after the deltoid-splitting incision (white arrowhead) and division of the acromioclavicular joint. (**B**) The supraspinatus tendon (white arrow) was sharply detached from the greater tuberosity, and the footprint (white arrowhead) was gently decorticated. (**C**) Crossed bone tunnels were drilled at the anterior and posterior margins of the footprint and 2 mm lateral to the articular surface using a 22-gauge needle. (**D**) The suture end was passed through the bone tunnels and firmly tied over the humeral metaphyseal cortex, anatomically repairing the supraspinatus tendon to its native footprint. (**E**) For the SIRT6 overexpression experiment, adenovirus SIRT6 (Ad-SIRT6) was injected into the repaired cuff tendon.

**Figure 2 cells-12-02035-f002:**
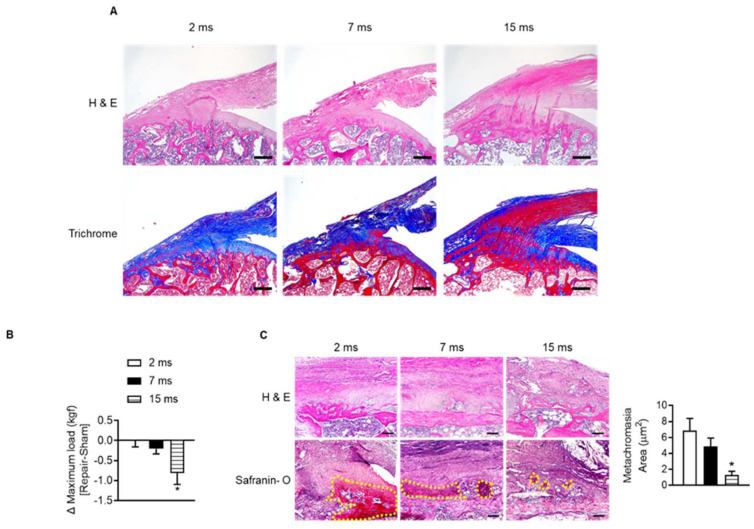
Impairment of tendon-to-bone healing after rotator cuff repair in aged rats. (**A**) Representative H&E staining and Masson’s trichrome staining of the supraspinatus tendon and proximal humeral head according to indicated age (scale bars = 250 μm). (**B**) Two-month-old, seven-month-old, and fifteen-month-old rats were subjected to sham (left side) or tenotomy and repair surgery (right side). The difference in maximum force (kilogram-force, kgf) between repair surgery after tenotomy (right side) and sham surgery (left side) in the same rat was analyzed four weeks after surgery (*n* = 4–5). * *p* < 0.05 versus 2 ms. (**C**) Representative H&E staining and Safranin O staining of the tendon-to-bone area after tenotomy with repair surgery (scale bars = 100 μm). Metachromasia areas (red-color-stained area: yellow dotted line) were analyzed at the tendon-to-bone site (*n* = 4), and the values are the mean ± SEM. * *p* < 0.05 versus 2 ms repair.

**Figure 3 cells-12-02035-f003:**
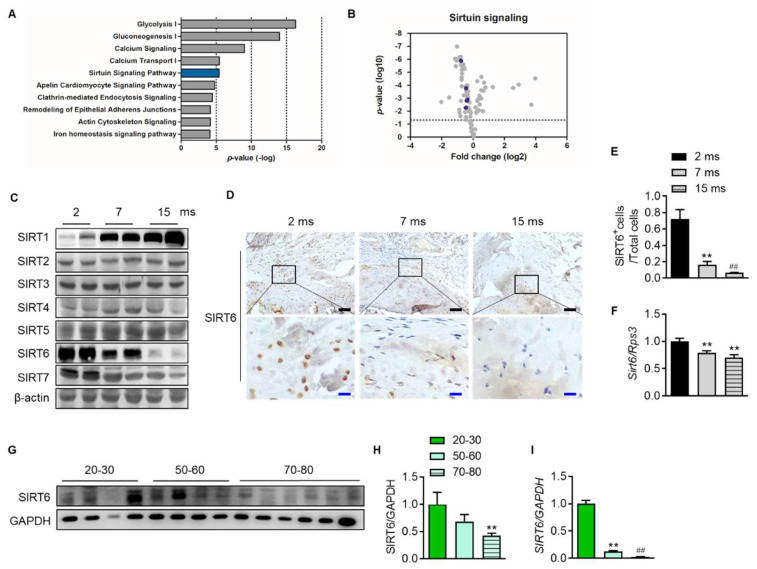
Attenuation of SIRT6 expression with age. (**A**) Canonical pathway enrichment analysis was calculated based on the protein expression fold change between the youngest-aged rats (2 ms, *n* = 4) and the oldest-aged rats (15 ms, *n* = 4) in this study. (**B**) Volcano plot of the pairwise comparison between the youngest-aged rat (2 ms) and the oldest-aged rat (15 ms) group proteomes. Expression fold changes were converted to log2 from the ratio value. The dotted line indicates the significance threshold (*p* < 0.05) by ANOVA statistical analysis, and the dark purple circle represents a protein related to the sirtuin signaling pathway. (**C**) Sirtuin protein levels were analyzed from the rotator cuff of rats aged 2 months, 7 months, and 15 months. (**D**) Representative immunohistochemical staining of SIRT6 in rotator cuff of rats. The bottom figures are enlarged figures of the black box above (black scale bars = 50 μm, blue scale bars = 10 μm). (**E**) Number of SIRT6-positive cells out of total cell count in the rotator cuff of rats (*n* = 3). (**F**) mRNA level of *Sirt6* from the rotator cuff of rats aged 2 months, 7 months, and 15 months (*n* = 6–8). (**G**,**H**) SIRT6 protein levels were analyzed from human subjects in their 20 s and 30 s, 50 s and 60 s, and 70 s and 80 s (*n* = 4–6). The band intensities were determined by densitometry and normalized to those of GAPDH. (**I**) mRNA level of *SIRT6* from the rotator cuff of human subjects in their 20 s and 30 s, 50 s and 60 s, and 70 s and 80 s (*n* = 4–6). ** *p* < 0.01 versus 20–30 s; ## *p* < 0.01 versus 50–60 s. The values are the mean ± SEM. ## *p* < 0.01 versus 7 ms; ** *p* < 0.01 versus 2 ms.

**Figure 4 cells-12-02035-f004:**
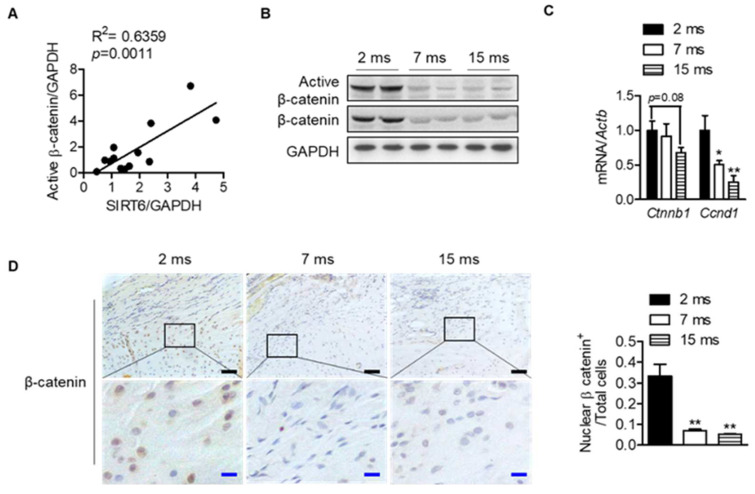
Decrease in Wnt-β-catenin signaling with age. (**A**) Scatter plots of SIRT6 protein expressions and active β-catenin protein expression from human subjects of the rotator cuff (*n* = 13). (**B**) Active β-catenin and total β-catenin expressions were analyzed from rotator cuff of rats aged 2 months, 7 months, and 15 months. (**C**) mRNA levels of Ctnnb1 and Cnnd1 according to age (*n* = 5–6). * *p* < 0.05 versus 2 ms; ** *p* < 0.01 versus 2 ms. (**D**) Representative immunohistochemical staining of β-catenin in rotator cuff of rats. The bottom figures are enlarged figures of the black box above (black scale bars = 50 μm, blue scale bars = 10 μm). Number of β-catenin-nuclear-positive cells out of total cell count in the rotator cuff of rats (*n* = 3). The values are the mean ± SEM. ** *p* < 0.01 versus 2 ms.

**Figure 5 cells-12-02035-f005:**
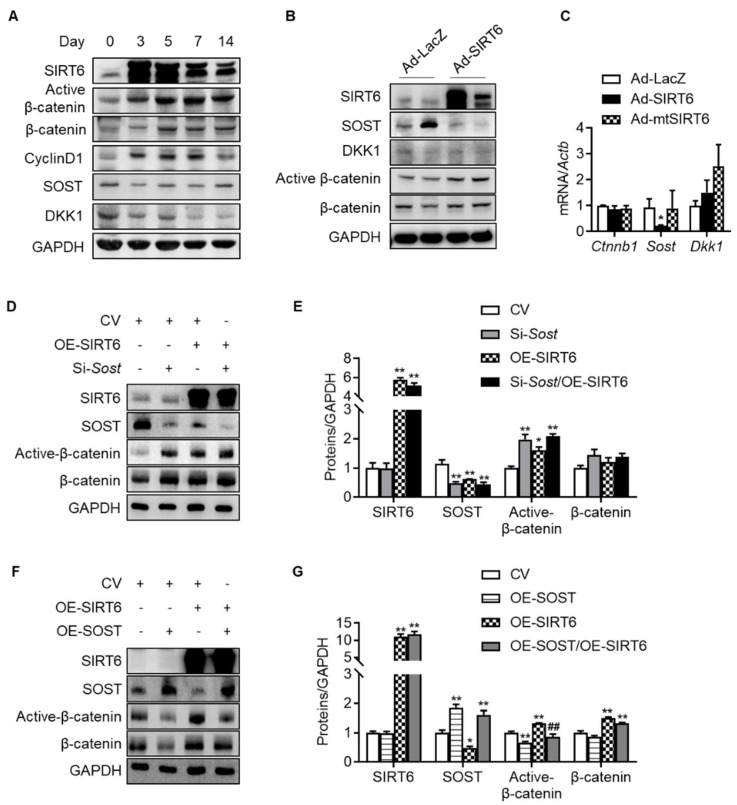
Regulation of Wnt-β-catenin signaling by SIRT6. (**A**) SIRT6 adenovirus was injected into the rotator cuff of aged rats (15 months) and harvested at the indicated time points. Proteins of β-catenin signaling and Wnt antagonists were examined using Western blotting. (**B**,**C**) Primary tenocytes from aged rats (15 months) were transduced with Ad-LacZ, Ad-SIRT6, and Ad-mutant SIRT6 (Ad-mtSIRT6). Proteins and mRNA levels for β-catenin signaling and Wnt antagonists were examined using Western blotting and real-time RT-PCR, respectively (*n* = 4). * *p* < 0.05 versus Ad-LacZ. (**D**–**G**) Protein levels of SIRT6, SOST, active β-catenin, and β-catenin after overexpression of SIRT6 with knock-down of SOST or with overexpression of SOST in primary tenocytes from aged rats (15 months). The band intensities were determined by densitometry and normalized to those of GAPDH (*n* = 4). CV; control vector, OE; overexpression. * *p* < 0.05 versus CV; ** *p* < 0.01 versus CV; ## *p* < 0.01 versus OE-SIRT6. The values are the mean ± SEM.

**Figure 6 cells-12-02035-f006:**
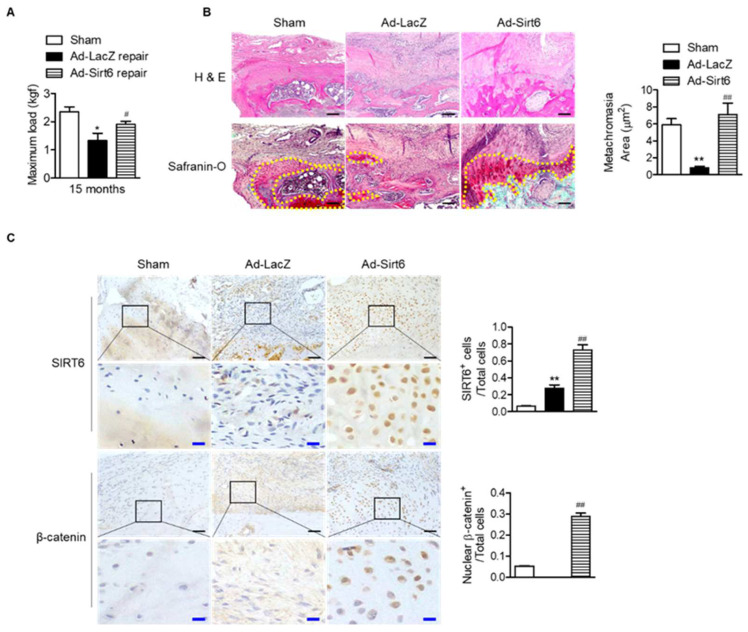
Improvement in tendon-to-bone healing by SIRT6 in aged rotator cuff. (**A**–**C**) Sirt6 adenovirus and LacZ virus were injected around rotator cuff area after rotator cuff tenotomy and repair in aged rats (15 months). (**A**) The maximum force (kilogram-force, kgf) between bone and tendon junction (*n* = 4–5). * *p* < 0.05 versus 15 ms sham; # *p* < 0.05 versus Ad-LacZ repair. (**B**,**C**) Representative H&E, Safranin O, and immunohistochemical staining in the rotator cuff insertion site of rats (black scale bars = 50 μm, blue scale bars = 10 μm). Metachromasia areas (red-color-stained area: yellow dotted line) were analyzed at the tendon-to-bone site (*n* = 4); number of SIRT6-positive cells or β-catenin-nuclear-positive cells out of total cell count (*n* = 4). The values are the mean ± SEM. ** *p* < 0.01 versus sham; ## *p* < 0.01 versus Ad-LacZ.

## Data Availability

The data presented in this study are available on request from the corresponding author.

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
