# Peer review of "Sirtuin 6 Overexpression Improves Rotator Cuff Tendon-to-Bone Healing in the Aged"

_cells, 2023, doi:10.3390/cells12162035_

Round 1

Reviewer 1 Report

In this study, the authors demonstrated that SIRT6 is an aging-associated protein in the rotator cuff and functions through Wnt signaling and that SIRT6 overexpression improves tendon-to-bone healing after acute injury in the aged rotator cuff. This study focused on the molecular mechanisms of aging-related rotator cuff tears, which have not been studied before and are of great significance. However, the following concerns should be addressed.

Major comments:

1. In the introduction, the biological process and significance of tendon-to-bone healing are better to be addressed.

2. Is there any previous research studying the relationship between sirtuins and Wnt signaling? If so, the content can be added to the introduction or discussion.

3. In method 2.2, the description of surgery is a bit difficult to understand, it would be better to provide a schematic diagram or pictures of the surgery process.

4. In Figures 2C, and D, the protein and mRNA levels of sirt6 in both rats and human are needed. And the original western blot figure of Figure 3A is also needed.

5. In results 2.5, there is no direct evidence showing sirt6 regulates β-catenin through SOST. It is better to observe the level of β-catenin by inhibiting SOST in the Ad-sirt6 group.

6. There is already sirt6 activator found in previous research(doi: 10.1038/s41589-018-0150-0.), why not try it on rats ?

Minor comments:

1. The description result in the abstract can be more concise.

Other 

The language of the article needs to be improved.

The language of the article needs to be improved.

Reviewer 2 Report

The authors did a nice study to establish one of the mechanisms involved in rotator cuff tears. The study was well-designed to answer their scientific question. The manuscript would need a minor revision before publication as listed below:

1.     Please verify if the study protocol has been approved by a related institution.

2.     Please mention the sample size calculation for both human and in-vivo studies in the “material and methods” section.

3.     Line 132: Please explain what was the area of interest to measure for histological analysis.

4.     Line 136: “Four rotator cuffs from 2-month-old and 15-month-old mice were isolated”. The reviewer just wondered did the authors use mice or rats in this study.

5.     It would be helpful if the authors could clarify how rt-qPCR and WB calculations were performed and normalized.

6.     Line 409-414: There are other limitations to using gene therapy methods. Please explains in the limitation paragraph.

Round 2

Reviewer 1 Report

Thanks to the authors for responding to my comments. The updates have addressed the majority of my concerns about the paper.

However, a few problems still exist:

1. For review 5, I’m sorry I didn’t mention that it’s also important to observe the level of β-catenin by overexpression SOST in the SIRT6-OE group. I would also like to see if the increase of β-catenin through SIRT6 overexpression can be reversed by the overexpression of SOST.

2. Quantitative analysis of western blotting are preferred in figure 5.

Extensive editing of English language required.
